# Persistence of *Xanthomonas campestris* pv. *campestris* in Field Soil in Central Europe

**DOI:** 10.3390/microorganisms9030591

**Published:** 2021-03-13

**Authors:** Filip Gazdik, Samuel Magnus, Steven J. Roberts, Rafal Baranski, Jana Cechova, Robert Pokluda, Ales Eichmeier, Dariusz Grzebelus, Miroslav Baranek

**Affiliations:** 1Mendeleum—Institute of Genetics, Mendel University in Brno, Valticka 334, 691 44 Lednice, Czech Republic; jana.cechova@mendelu.cz (J.C.); ales.eichmeier@mendelu.cz (A.E.); dariusz.grzebelus@urk.edu.pl (D.G.); miroslav.baranek@mendelu.cz (M.B.); 2Department of Fruit Science, Mendel University in Brno, Valticka 337, 691 44 Lednice, Czech Republic; xmagnus@node.mendelu.cz; 3Plant Health Solutions Ltd., 20 Beauchamp Road, Warwick CV34 5NU, UK; s.roberts@planthealth.co.uk; 4Department of Plant Biology and Biotechnology, University of Agriculture in Krakow, AL. 29 Listopada 54, 31-425 Krakow, Poland; rafal.baranski@urk.edu.pl; 5Department of Vegetable Science and Floriculture, Mendel University in Brno, Valticka 337, 691 44 Lednice, Czech Republic; robert.pokluda@mendelu.cz

**Keywords:** brassicas, *Xanthomonas campestris* pv. *campestris*, persistence, soil, detection, PCR, nested real-time PCR

## Abstract

*Xanthomonas campestris* pv. *campestris* (*Xcc*) is a bacterium that causes black rot of crucifers. The greatest losses of brassica crop production usually result from seed-borne infection, but carry-over of inoculum in field soil may also be possible. The aim of this study was to monitor persistence of *Xcc* in field soil in central Europe using a conventional PCR assay with hrpF primers and a two-step nested real-time PCR assay using Zur primers. The work has demonstrated that nested real-time PCR can be used to improve the analytical sensitivity for detection of *Xcc* in soil compared to conventional PCR, and that *Xcc* may persist in soil for up to two years following an infected brassica crop in central European climatic conditions.

## 1. Introduction

*Xanthomonas campestris* pv. *campestris* (Pammel) Dowson (*Xcc*), causes black rot of crucifers [1] and has a worldwide distribution [2]. It is considered one of the most important diseases of brassicas [1,3] and may result in significant losses in brassica crop production [2]. Recent reviews on *Xcc* can be found in Mansfield et al. [4] and Vicente & Holub [5].

Although it is considered to be primarily seed-borne [6,7], other potential sources of inoculum include weeds [8], crop debris and soil [9,10]. *Xcc* has been reported to survive in plant debris in soil for up to two years in the Netherlands [11], but there is no data for central Europe.

Various studies have been made regarding the persistence of *Xcc* in soil, but most of them have focused on survival in association with crop debris. Thus it has been well established that *Xcc* can potentially survive in brassica crop debris in soil as long as debris remains visibly present [9,11,12,13,14,15]. On the other hand, Schaad & White [9] suggests that *Xcc* cannot survive in soil for more than 42 days in winter and 14 days in summer without the host plant debris.

Most of the studies have used dilution plating assays for detection of the pathogen. As an alternative to traditional plating methods for detection of *Xcc* in seeds and planting material [16], a PCR method targeting the *hypersensitivity reaction and pathogenicity* (*hrpF*) gene was developed by Berg et al. [17]. This gene is highly conserved within the *Xanthomonas campestris* species. The hrpF primer pair targets the 3’ end of the *hrpF* gene and amplifies a 619 bp product.

Another recently developed method for detection of *Xcc* in cabbage tissues and seeds is a nested real-time PCR assay targeting *Zur* (*Zinc uptake regulator*) gene [18]. A pair of outer primers amplify a 305 bp product in an initial reaction and an internal region is then amplified in real-time PCR using a second pair of primers.

In this paper we monitor the persistence of *Xcc* in a field soil using two different assays, the conventional PCR assay using hrpF primers [17] and the recently developed nested real-time PCR assay [18] for detection of *Xcc* DNA in a field soil under natural conditions for two years following a black rot infected cabbage crop in the Czech Republic.

## 2. Materials and Methods

### 2.1. Experimental Site and Sampling

Seeds of a head cabbage (*Brassica oleracea* var. *capitata* L., cv. Albatros F1) were inoculated with *Xcc* strain 1279A (WHRI—Warwick Horticulture Research International, Warwick, UK), race 4 [19], by immersing in a bacterial suspension containing 10^8^ CFU·mL^−1^ for 1 h. The seeds were then drained and spread in a thin layer in shallow trays on paper towels and left to dry overnight at room temperature in the air-flow of a fume hood. Seeds treated by immersing in a physiological solution were used as a negative control. The seeds were then directly sown on a 4.5 × 10 m square experimental field that was part of a different experiment at the Faculty of Horticulture in Lednice, part of Mendel University in Brno (Czech Republic) on 14 May 2016. After harvesting on 14 October 2016, all visible plant debris was removed. An area 2 × 2 m square (25 infected cabbage plants were growing in this area) within the field was demarcated for sampling and remained uncultivated (but not weed free) for the duration of the experiment. The surrounding area was regularly mulched with a tractor mounted mulcher and kept uncultivated for the duration of the experiment. Soil sampled around the demarcated area was used as a negative control.

Soil samples were collected from nine fixed sampling locations (Figure 1) within the demarcated area at intervals of approximately two months over the following two years. Approximately 25 g of soil was collected from each of the sampling locations, using an open-sided column soil cylinder (10 mm in diameter), from a depth of 5 to 10 cm and without any visible admixtures e.g., plant parts, stones, insects. The soil cylinder was sterilized with 96% ethanol after each sample. The samples were then stored in aluminium Ziploc^®^ bags in a refrigerator up to one week until processing. A total of 108 soil samples (9 samples from each of 12 samplings) were collected from 12 December 2016 to 22 September 2018.

### 2.2. Analysis of Soil Organic Matter

Soil organic matter (SOM) and soil oxidizable carbon (SOC) were determined from the soil samples by Walkley-Black method [20], modified by Novak-Pelisek [21]. The soil organic matter was calculated using the equation SOM (%) = SOC (%) × 1.724 as described by Jandak et al. [22]. The total SOC was evaluated according to Zaujec et al. [23] and soil quality based on SOM content was evaluated according to Jandak et al. [22].

### 2.3. DNA Extraction

Microbial genomic DNA was extracted from soil samples using a DNeasy^®^ PowerSoil^®^ Kit (Qiagen, Hilden, Germany) according to the user manual except for a final elution volume of 50 µL instead of 100 µL. Each 25 g soil sample was mixed thoroughly and a 0.25 g sub-sample (not dried or sieved) was used for extraction. A soil sample from negative variants of the previous experiment was used as negative control and a sample from the same soil with a *Xcc* suspension added prior to extraction was used as a positive control in each assay. The concentrations of DNA in the extracts were measured with ModulusTM (TurnerBiosystems, Sunnyvale, CA, USA) using a PicoGreen^®^ kit (Molecular Probes, Eugene, OR, USA), according to the user manual.

### 2.4. Conventional PCR Detection of the hrpF Gene

Primer pair DLH 120 and DLH 125 (Table 1) designed by Berg et al. [17] were used to amplify a 619 bp fragment of *hrpF* gene region. For specific amplification of the *hrpF* region, a GoTaq^®^ G2 Flexi kit (Promega, Madison, WI, USA) was used as described in Eichmeier et al. [18]. The reaction mixture contained 10.5 µL of nuclease-free water (Ambion, Foster City, CA, USA), 4 µL 5× for polymerase, 0.7 µL 5× Green GoTaq^®^ G2 Flexi Buffer (Promega, Madison, WI, USA), 0.2 µL GoTaq^®^ G2 Flexi DNA polymerase (5 U·µL^−1^) (Promega, Madison, WI, USA), 1.2 µL MgCl_2_ (25 mM) (Promega, Madison, WI, USA), 0.2 µL of dNTPs mix (10 mM) (MCLAB, San Francisco, SF, USA), 1 µL of each primer (10 µM) and 2 µL of template DNA, total volume 20.8 µL. Amplification was performed on a Professional Gradient thermal cycler (Biometra, Göttingen, Germany), with the following temperature profile: initial denaturation step at 95 °C for 3 min, followed by 40 cycles of 95 °C for 40 s, 60 °C for 40 s and 72 °C for 40 s, with the final step at 72 °C for 7 min. PCR products were separated on an agarose gel (1.2%), containing 0.005% GelRedTM (Biotium, Fremont, CA, USA) and visualized on a UV transilluminator. The presence of a distinct band of approximately 619 bp was taken to indicate the presence of *Xcc* DNA in the template. PCR products were sequenced according to Eichmeier et al. [24] in order to validate the amplicon sequences.

### 2.5. Nested Real-Time PCR Detection of the Zur Gene

#### 2.5.1. Pre-Amplification

For pre-amplification, primer pair Zur1-EAC-fwd and Zur1-CAE-rev [25] (Table 1) were used, targeting the *Zinc uptake regulator* (*Zur*) gene, with product length of 305 bp. The reaction mixture was the same as previously described for the hrpF fragment. Amplification was performed on a TProfessional Gradient thermal cycler, with the following temperature profile: an initial denaturation step at 95 °C for 3 min, followed by an optimized number of 20 cycles of 95 °C for 40 s, 51 °C for 40 s, and 72 °C for 40 s. The PCR mixture obtained within the PCR pre-amplification was subsequently used as a template for the real-time PCR.

#### 2.5.2. Quantitative Real-Time PCR

Primer pair Zur2-EAC-fwd and Zur1-CAE-rev, and a Zur1-TP probe, designed by Eichmeier et al. [18,25] (Table 1), were used for specific amplification of a short region of the *Zur* gene using a TaqMan^®^ Real Time PCR system (MCLAB, San Francisco, SF, USA). The targeted gene region lies within the region targeted by the primer pair used in the pre-amplification (Zur1-EAC-fwd and Zur1-CAE-rev) with a product length of 142 bp. The reaction contained 6 µL of nuclease-free water, 10 µL of 2× HoTaq Real-Time PCR Kit (MCLAB, San Francisco, SF, USA), 0.8 µL of both Zur2-EAC-fwd and Zur1-CAE-rev primers (10 µM), 0.4 µL of TaqMan^®^ Zur1-TP probe (10 µM) (MCLAB, San Francisco, SF, USA) and 2 µL of the template DNA (pre-amplified), total volume 20 µL. Amplification was performed in a 72-well Rotor-GeneTM3000 thermal cycler (Rotor-Gene^®^, QIAGEN, Hilden, Germany), with 0.1 mL Optical 4-tube-strips (Rotor-Gene^®^, QIAGEN, Hilden, Germany), with the following temperature profile: initial denaturation step at 95 °C for 3 min, followed by 35 cycles of 95 °C for 40 s, 54 °C for 40 s, and 72 °C for 40 s, with the final step at 72 °C for 7 min. PCR products were sequenced according to Eichmeier et al. [24] in order to validate the amplicon sequences.

### 2.6. Real-Time PCR Standards Preparation

DNA standards for preparation of a calibration curve were generated by cloning of PCR product of *Xcc* (WHRI 1279A) into a pCR^™^4-TOPO^™^ Vector (Invitrogen, Carlsbad, SD, USA). The length of insert was 142 bp. Subsequently, a 100 µL standard of known number of copies (10^7^ copies·µL^−1^) was prepared (Generi Biotech, Hradec Kralove, Czech Republic).

### 2.7. Detection Limits and Sensitivity

In order to determine the detection limits of the two assays a three-day old culture of *Xcc* 1279A growing on Mueller-Hinton medium (HiMedia, Mumbai, India) was suspended in 0.85% saline and a ten-fold dilution series prepared (also in saline). Aliquots (1 mL) of each dilution were centrifuged for 1 min at 12,000× *g*, most of the supernatant removed, the pellet resuspended and the entire residual volume then added to 0.25 g sub-sample of soil which had been previously tested and given negative results in both PCR assays. The DNA was extracted from the prepared samples and assayed by both PCR methods as described above. The number of *Xcc* in the suspension was determined by dilution plating on plate count agar (HiMedia, Mumbai, India).

### 2.8. Data Analysis

Data were analyzed by two different approaches using the statistical analysis software R, version 3.4.4 (R Core Team, Vienna, Austria).

#### 2.8.1. Inverse Estimation of *Xcc* from Nested Real-Time PCR Ct Values

In the first approach, data from the nested real-time PCR were analyzed by interpreting the Ct values as being directly proportional to numbers of *Xcc* in the tube according to the soil calibration curve, using inverse estimation. A linear regression of log_10_ (CFU) *Xcc* per tube was fitted to the Ct values using the lm() function in R. Estimates of the log_10_ (CFU) of *Xcc* per tube were then obtained as predictions from the model using the predict() function. The log values were then back-transformed to CFU, and values less than 1 were set to 0 (on the basis that we cannot detect less than one bacterial cell per tube). A generalized linear model was then fitted to the bacterial numbers using the glm() function with a quasi-Poisson error distribution and log link function. The sample date was used as a classifying factor, and the log of the equivalent weight of soil represented in the reaction tube was fitted as an offset.

#### 2.8.2. Most Probable Numbers (MPNs)

In the second approach, the data from both the conventional PCR and the nested real-time PCR were analyzed as binomial values (i.e., detection or not of *Xcc* DNA in the PCR tube). The combined data for the nine samples at each sampling date were then used to estimate the mean population of *Xcc* in the soil as most probable numbers (MPNs) [26]. Estimates were obtained by fitting a generalized linear model to the data using the glm() function with binomial error distribution and complementary log-log link function using the sample date as a classifying factor, and using the log of the equivalent weight of soil in the reaction tube as an offset. Where all samples were either positive or negative, confidence limits were obtained using the mpn() function.

#### 2.8.3. Weather Data

Daily weather data were obtained from a meteorological station located within the Faculty of Horticulture in Lednice, approximately 40 m away from the trial site. The data were summarized as means over various time periods (1, 2, 4, and 8 weeks) prior to sample collection. A preliminary investigation of the relationships amongst the weather variables and the estimates of *Xcc* was done by generating a correlation matrix of all pairwise correlations using the Spearman correlation coefficient.

## 3. Results

Prior to harvesting, all 25 plants in the experimental area showed typical symptoms of *Xcc* infection (yellow V-shaped lesions with blackened veins).

A range of weeds grew in the experimental plot during the experiment consisted mainly of *Cirsium arvense* (L.) Scop., *Echinochloa crus*-*galli* (L.) P. B., *Papaver rhoeas* L. and *Bromus hordaeceus* L., with smaller amounts of *Capsella bursa*-*pastoris* (L.) Med., *Amaranthus retroflexus* (L.) and *Portulaca oleracea* (L.).

### 3.1. Detection Limits and Sensitivity

The DNA concentration in the soil extracts ranged from 4 to 243 ng·mL^−1^, but there was no consistent relationship with the positivity by either conventional or nested real-time PCR. In the conventional PCR, soil samples containing ≥ 1.1 × 10^3^
*Xcc* CFU·g^−1^ of soil produced a visible band (Figure A1 and Figure A2). The nested PCR was about 10 times more sensitive and produced a detectable PCR product (Figure A3) from soil samples containing ≥ 1.6 × 10^2^ CFU·g^−1^ of soil, with a linear relationship between the Ct value and the numbers of *Xcc* over the range examined (Figure 2).

### 3.2. Estimation of Xcc by Inverse Estimation Using Nested Real-Time PCR Ct Values

The calibration data (Ct vs. log_10_ CFU *Xcc* per PCR tube) (Figure 2) had an excellent linear fit (r^2^ = 0.99, *P* < 0.001) with a slope of −3.64, indicating a PCR efficiency of 88%. The intercept was 23.8 when log_10_ (CFU) = 0 (i.e., CFU of *Xcc* in the PCR tube = 1). As it makes no sense to be able to detect less than a single bacterial cell in a tube, and the *Zur* gene copy number per cell has been estimated as one [25], we therefore took this value to be the cutoff for detection of *Xcc* based on the real-time PCR (Table A1).

The numbers of *Xcc* detected per tube were estimated using the “inverse estimator” approach. As the underlying measure of *Xcc* in the soil is a count, a priori it would be expected to follow a Poisson distribution. Initial fitting of a generalized linear model with Poisson error distribution and sampling date as a fixed factor indicated that the data were over-dispersed, therefore the model was refitted with a quasi-Poisson distribution. Examination of the model coefficients and their standard errors indicated a significant intercept term (*P* < 0.001) but no effect of sampling date. Means and standard errors for each sampling date were obtained as predictions from the model, and are plotted against time in Figure 3A together with the 95% confidence limits.

Further investigation of the residuals for these data, indicated that three data points had high leverage, corresponding to a single low Ct value of approximately 11 at each of the sample dates where numbers apparently peaked (sample dates 2, 4 and 7).

### 3.3. Most Probable Numbers (MPNs)

The most probable number estimates and their 95% confidence limits for *Xcc* in the soil derived from the nested real-time PCR and conventional PCR are shown in Figure 3B,C, respectively. The values derived from conventional PCR in Figure 3C were adjusted for the higher detection limit. It should be noted that where all samples are positive or negative it is not possible to obtain an estimate, only a lower or upper confidence limit. The values of the MPN estimates were generally lower than those obtained by inverse estimation (IE) from the Ct-derived values. Also, compared to the IE Ct derived values, the MPNs were more precise (had narrower confidence limits) for estimable values. Nevertheless, examination of the model coefficients and their standard errors indicated that differences in numbers of *Xcc* between sample dates were not significant.

### 3.4. Associations between Estimated Numbers of Xcc and Weather Variables

As the numbers of *Xcc* do not follow a normal distribution, the Spearman coefficient was used to generate pairwise correlations between the estimates of *Xcc* and weather variables (air temperature, soil temperature at different depths, precipitation, relative humidity) summarized over periods of 1, 2, 4, and 8 weeks prior to sampling. Inevitably many of the weather variables (and especially temperature) were highly inter-correlated. In the region where the trial was conducted there is a tendency for winters to be cold and dry with most precipitation falling during the summer months, and this was reflected in positive correlations between temperature variables and precipitation.

The five most significant correlations between weather variables and each of the *Xcc* estimates are shown in Table 2.

For the IE Ct derived values and the MPN values from conventional PCR there were no significant correlations. In the case of MPN values from the nested real-time PCR there were significant negative correlations with many temperature variables, the strongest being with the mean maximum air temperature in the week prior to sampling (r = −0.69, *P* < 0.05). This relationship is shown in Figure 4.

### 3.5. Soil Organic Matter Content

There was no evidence of any systematic variation in the organic matter in the soil samples, values ranged from 1.3 to 2.3% with a mean of 1.7% (Table A2, Figure A4).

## 4. Discussion

Taken together, these results indicate that low levels of *Xcc* (or at least its DNA) may persist in field soil in the Czech Republic for up to two years following an infected cabbage crop.

We cannot be certain that the *Xcc* DNA detected represents viable bacteria. Several studies have shown that extracellular bacterial DNA can persist in soil for significant periods of time [27,28,29,30], and the rate of degradation is likely to be dependent on factors such as the soil type, temperature, moisture, organic matter and their impact on biological activity. Nevertheless, given the duration of the experiment and the two month intervals between sample dates, it would seem likely that most of the DNA detected was derived from cellular (viable) *Xcc*.

There have been a number of other studies that have examined the survival of *Xcc* in the soil environment. Most studies have focused on survival in association with crop debris and it has been well established that *Xcc* can potentially survive in brassica crop debris in soil as long as the debris remains visibly present (up to two years in the case of tough stem residues) [9,11,12,13,14,15]. On the other hand, when planktonic *Xcc* has been added to soil, survival time has been relatively short, e.g., up to six weeks in winter [9] and less at warmer temperatures. It is also clear that survival in both planktonic state and in debris is related to temperature, e.g., Kocks et al. [31] found no decline in numbers over 20 weeks at 0 °C, but a decline to almost undetectable levels at 20 °C in the same soil, with the same inoculum. Similarly Silva Júnior et al. [32] also found that survival declined with increasing temperatures.

Our results are consistent with the previous studies showing survival of *Xcc* in soil in association with crop debris for up to two years, but unlike other studies we did not intentionally bury infected debris in the soil. However, although visible crop debris was removed from soil surface at the beginning of the experiment, this does not preclude the possibility that *Xcc* was protected in crop residues remaining in the soil.

We have been assuming the *Xcc* detected has been surviving in the soil from an initial population derived from the infected crop that was harvested in October 2016. However, given that the experimental plot was left fallow and weeds were allowed to grow, the occasional occurrence of the cruciferous weed shepherd’s purse (*Capsella bursa*-*pastoris* L.), which is known to be susceptible to *Xcc* race 4 [33], could also have affected the dynamics of detectable *Xcc* in the soil, in addition to initial population resulting from the infected crop.

It is not possible to say if the relatively low numbers of *Xcc* we detected in soil would have any epidemiological significance if the field was replanted with another susceptible crop. Just because inoculum is present in the soil does not mean it will necessarily reach the target host, and even then, given the relatively low numbers present, the likelihood of infection would be low. Kocks et al. [31] replanted infected plots with healthy seedlings in three successive years, observing only a single infected plant out of the 588 planted over the three years.

Analysis of the data was problematic, and it was initially tempting to ascribe significance and seek explanations for the apparent variation in numbers from one sampling date to another. For example: a steady decline in the first summer, an increase in the winter and then another decline the following summer, and then an increase in late summer. However, the analysis indicated that differences between sampling dates were not significant. Whilst the absence of a significant difference does not necessarily mean that the observed differences are not real, it does highlight that the results are very variable (over-dispersed), and any apparent trends should be interpreted with great caution.

Thus, it is possible that there may have been a stable level of *Xcc* (DNA) in the soil but there was variability in the extraction and detection efficiency between samples and from sampling date to date. It is very unlikely that *Xcc* is randomly distributed in the soil, and so alternatively the variability observed may have been due to spatial heterogeneity in the levels of *Xcc* (DNA) in the soil, perhaps due to pockets of relatively large populations in some samples and none in most others, e.g., if they were protected in (micro) debris or clumps of cells. This would be consistent with the observations of apparent outliers in the Ct data at sample dates 2, 4, and 7. In practice, both variability in extraction and detection and spatial heterogeneity are likely to have contributed to the overall variability observed.

The negative correlation between *Xcc* (MPNs from nested real-time PCR) and temperature preceding sample collection (Table 2, Figure 4) suggests that *Xcc* numbers may be higher when temperatures are lower. This may seem counter-intuitive, as *Xcc* is considered a warm temperature pathogen with an optimum growth temperature of around 28 °C in vitro [34]. It therefore seems more likely that when soil temperatures are low, *Xcc* was easier to detect (e.g., if overall biological activity is low during the winter months, and the ground may be frozen, extraction and detection of relatively low abundance DNA may be easier), rather than due to an increase in numbers of *Xcc*.

In this study we compared two different PCR assays for detection of *Xcc* DNA from the same soil extracts: a conventional single-step PCR using the hrpF primers [17] and a nested real-time PCR using Zur primers [18,25]. Nested real-time PCR is considered as potentially being more sensitive than conventional single-step PCR [35,36,37,38], and this was also the case in this work, with the nested real-time PCR having an analytical sensitivity approximately 10 times greater than the conventional PCR, with more ‘positive’ results. It should also be noted that PCR is only the final step in the overall assay, and thus while the PCR itself may be able to detect DNA equivalent to a single bacterial cell in a PCR tube, this represents only a small fraction of the original sample (in this case 0.01 g of soil), hence the overall detection limit of the nested PCR assay was 300 CFU·g^−1^ (the upper confidence limit of a negative result, *P* = 0.95). Given the sensitivity of PCR we cannot be certain that all these positive results represent ‘true positives’, and it appears that the nested real-time PCR also introduced more variability into the results (presumably due to factors such as the 60% increase in cycle number, and increased number of operations e.g., pipetting), making direct interpretation of results more challenging. The precision of estimates was apparently improved by taking an MPN approach to interpretation of the data, and, with better overall assay design, e.g., by inclusion of multiple dilutions, or an efficient sequential assay design [39] this precision has the potential for further improvement.

## 5. Conclusions

This work has demonstrated that *Xcc* may persist in soil for up to two years following an infected brassica crop in central European climatic conditions and that the nested real-time PCR can be used to improve the analytical sensitivity for detection of *Xcc* in soil compared to conventional PCR. Whether this has any significance for the epidemiology and control of brassica black rot caused by *Xcc* remains unanswered and will require further work to estimate the risk of infection in a subsequent crop from the relatively low levels *Xcc* detected.

## Figures and Tables

**Figure 1 microorganisms-09-00591-f001:**
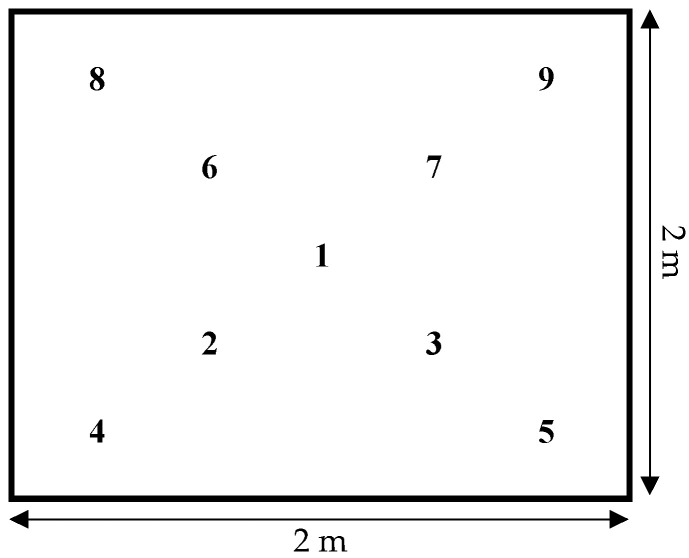
Scheme of experimental plot.

**Figure 2 microorganisms-09-00591-f002:**
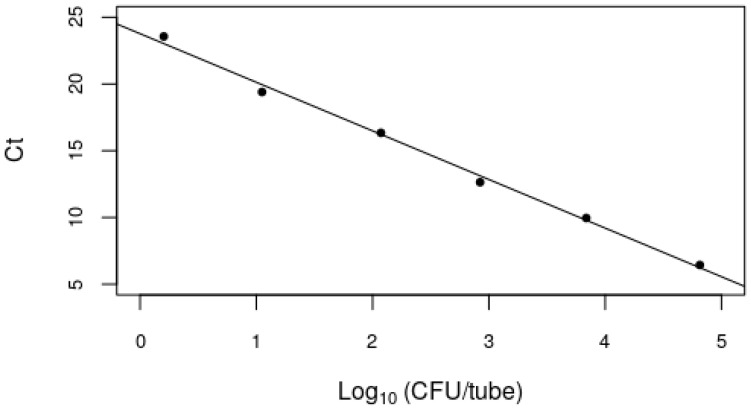
Calibration curve of nested real-time PCR assay using the Zur1-CAE-rev, Zur2-EAC-fwd primer pair and Zur1-TP probe (Table 1) and dilution of *Xcc* added to soil.

**Figure 3 microorganisms-09-00591-f003:**
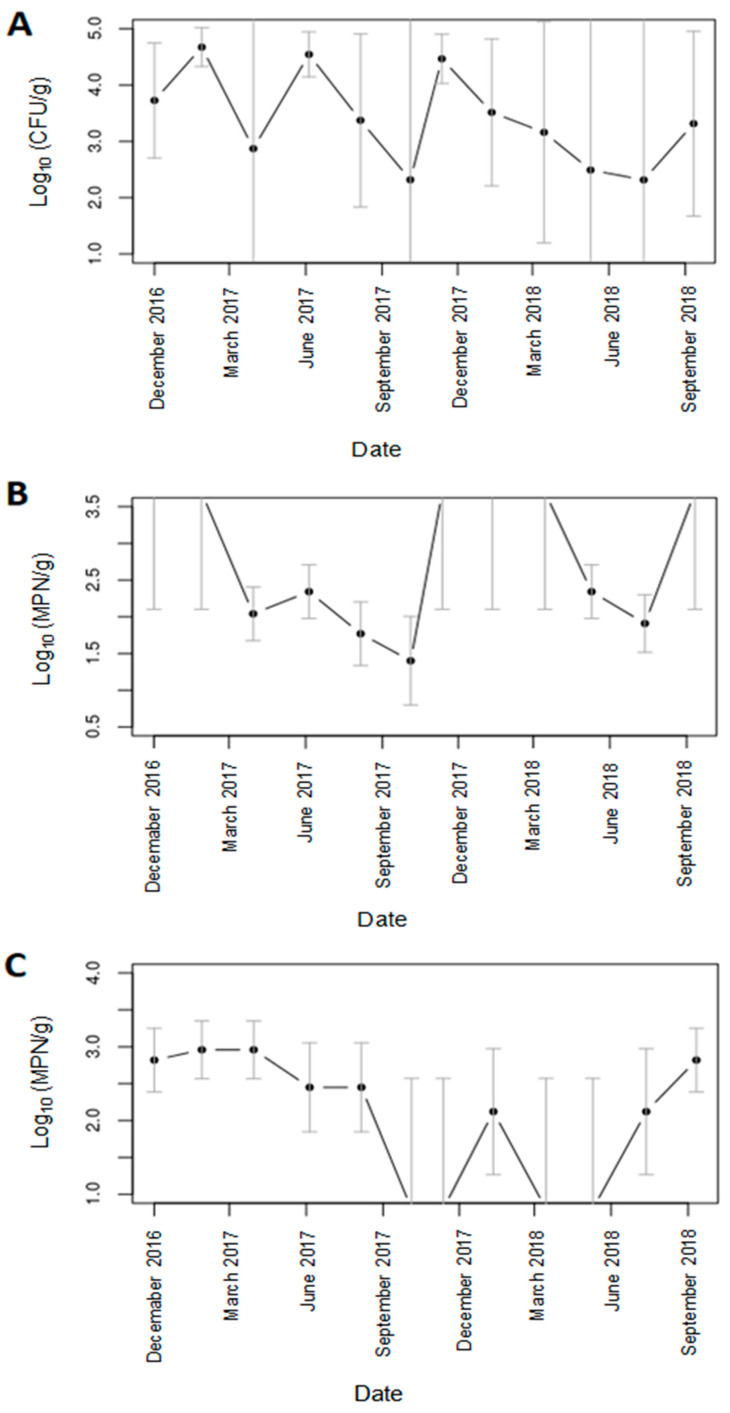
Estimated mean numbers of *Xcc* in soil obtained by: (**A**) inverse estimation from nested real-time PCR Ct values (**B**) most probable numbers from nested real-time PCR (**C**) most probable numbers from conventional PCR with hrpF primers. Values were obtained as predictions from the appropriate generalized linear model. Error bars represent the 95% confidence intervals. It should be noted that values outside the scale range in (**B**,**C**) represent sample dates when all samples were positive (**B**) or al samples were negative (**C**) and therefore it is not possible to obtain estimates, only lower (**B**) or upper (**C**) confidence limits.

**Figure 4 microorganisms-09-00591-f004:**
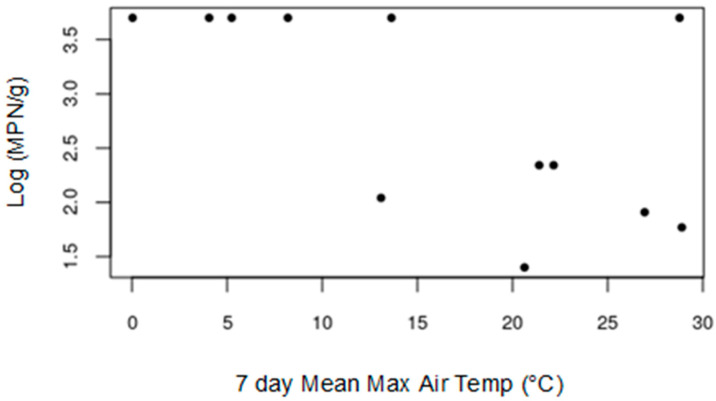
Scatter plot showing relationship between the log(MPN) *Xcc* detected derived from nested real-time PCR and the mean maximum air temperature in the week prior to sampling.

**Table 1 microorganisms-09-00591-t001:** List of used primers with their sequences, targeted genes and product sizes.

Primer	Sequence (5′–3′)	Targeted Gene	Product (bp)
DLH120 [17]	CCGTAGCACTTAGTGCAATG	*Hypersensitivity Reaction and Pathogenicity gene* (*hrpF*)	619
DLH125 [17]	GCATTTCCATCGGTCACGATTG		
Zur1-CAE-rev [25]	AGGCGACGAAGGCATTGA	*Zinc Uptake Regulator gene* (*Zur*)	305
Zur1-EAC-fwd [25]	AACGCACGACCAGGAACA		
Zur2-EAC-fwd [18]	CAAACCGGTCAAGGCCTA	*Zinc Uptake Regulator gene* (*Zur*)	142
Zur1-CAE-rev [25]	AGGCGACGAAGGCATTGA		
Zur1-TP [18]	FAM-CGCTGGATTTTTTGATGGBHQ		

**Table 2 microorganisms-09-00591-t002:** The top five most significant correlations (**r**) between each of the *Xcc* estimates and means of weather variables, and their significance (***P***). The values in parentheses represents the time period over which the mean was calculated. IE, inverse estimation; MPN, most probable number.

	r	*P*
**IE from Ct**		
Precipitation (28 days)	−0.49	0.106
Mean Air Temp. (7 days)	−0.48	0.118
Min Air Temp. (14 days)	−0.45	0.145
Soil Temp. at 20 cm (7 days)	−0.43	0.159
Relat. Humidity (56 days)	0.43	0.159
**MPN Real-Time PCR**		
Max. Air Temp. (7 days)	−0.69	0.012
Soil Temp. at 20 cm (7 days)	−0.69	0.012
Min. Ground Temp. (14 days)	−0.69	0.012
Mean Air Temp. (28 days)	−0.69	0.012
Max. Air Temp. (28 days)	−0.69	0.012
**MPN *hrpF***		
Precipitation (14 days)	−0.41	0.182
Precipitation (7 days)	0.32	0.306
Sunshine hours (14 days)	0.27	0.402
Precipitation (56 days)	−0.18	0.58
Mean Air Temp. (7 days)	−0.15	0.643

## Data Availability

All the data are included in this paper and in Appendix A.

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
