# Peer review of "Persistence of Xanthomonas campestris pv. campestris in Field Soil in Central Europe"

_microorganisms, 2021, doi:10.3390/microorganisms9030591_

Round 1
Reviewer 1 Report
This study aimed to detect Xanthomonas campestris pv. campestris, responsible for black rot in cabbage, in the field soil, using PCR assays. The outcome of this study is critically important for the European region as it is impacting cabbage cultivation. This study also provided the PCR tool to determine the epidemiology of black rot diseases. This manuscript is missing some of the methodologies that are significant for end-users (industries, diagnostics lab). In my opinion, this is a technically important article, and materials and methods are critically important, as it is one of the major outcomes of this study. As such, I suggest to include the below-mentioned methods.
Line 60-61: Xcc – explain where is this culture obtained?
Line 63-67: Please justify why control treatment is not included for this study.
Line 79-81: Explain the soil processing procedure ( 108 soil samples). Soil DNA extraction is complex process, specify - is this samples were dried prior to extraction to avoid moisture - is it sieved to increase the DNA extraction efficiency?
Line 88: What is SOC?
Line 100: How DNA quantity was taken for PCR assay?
Line 100: there was no internal control used in the DNA extraction - why? how to justify false positive/false negatives?
Line 170: In this study, PCR product/qPCR products were not sequenced to confirm the amplification. However, it is necessary as these primers were adapted from a previous study.
Line 190-201: This is not necessary for this study, may be worth considering in the future study Gibbs and Gower method is an alternative option for quantifying the pathogen incidence in the seed or soil. (link : https://doi.org/10.1094/PDIS-03-18-0503-RE).
Author Response
Question: line 60-61 – Xcc – explain where is this culture obtained?
Answer: new line 77-78 – Origin of the Xcc strain 1279A was added as WHRI to the sentence “…were inoculated with Xcc strain 1279A (WHRI Warwick Horticulture Research International, UK), race 4 [19] by immersing in…”
Question: line 63-67 – Please justify, why control treatment is not included for this study.
Answer: new line 81 – Sentence “Seeds treated by immersing in a physiological solution were used as a negative control” was added. The experiment where an infected crop was grown was part of previous study, where various variants were inoculated with Xcc strain 1279A and a negative control was also included. The current experiment was established on an area, where positive variants from the previous experiment were grown and soil from surrounding area negative to Xcc was used as negative control (see Figure A1 – N-negative control). This negative control was included in each of the conventional and nested real-time PCR runs.
Question: line 79-81 – Explain the soil processing procedure (108 soil samples). Soil DNA extraction is complex process, specify – if this samples were dried prior to extraction to avoid moisture – is it sieved to increase the DNA extraction efficiency?
Answer: new line 98-99 – Part of sentence “A total of 108 soil samples were collected…” was changed to “A total of 108 soil samples (9 samples from each of 12 samplings) were collected…”. Line 130 – Sentence “…0.25 g sub-sample was used for extraction.” was changed to “…0.25 g sub-sample (not dried or sieved) was used for extraction.”. The soil extraction procedure was done using a specialized kit according to the user manual, where “raw” soil is just mixed thoroughly and directly used for the extraction without drying or sieving.
Question: line 88 – What is SOC?
Answer: new line 119-123 – Sentence “The soil organic matter was calculated using the equation SOM (%) = SOC (%) × 1.724 as described by Jandak et al. [19], where SOc represents soil oxidizable carbon.” was restructured as “Soil organic matter (SOM) and soil oxidizable carbon (SOC) were determined from the soil samples by Walkley-Black method [20], modified by Novak-Pelisek [21]. The soil organic matter was calculated using the equation SOM (%) = SOC (%) × 1.724 as described by Jandak et al. [22].”.
Question: line 100 – How DNA quality was taken for PCR assay?
Answer: The concentration of DNA was not adjusted in any way. Original concentration was used for all analyses. New lines 281-283 – added sentence “The DNA concentration in the soil extracts ranged from 4 to 243 ng·ml-1, but there was no consistent relationship with the positivity by either conventional or nested real-time PCR.”.
Question: line 100 – There was no internal control used in the DNA extraction – why? How to justify false positive / false negatives?
Answer: new line 130-133 – Added sentence “Soil sample from negative variants was used as negative control and simultaneously sample from the same soil with added Xcc suspension prior to extraction was used as a positive control in each assay.” During the DNA extraction, both negative and positive controls were included. Negative control was soil sample taken from the surrounding area where negative controls from the previous experiment were grown (negative to Xcc) and positive control was sample from same negative soil with added Xcc suspension directly into the sample prior to extraction.
Question: line 170 – In this study, PCR product/qPCR products were not sequenced to confirm the amplification. However, it is necessary as these primers were adapted from a previous study.
Answer: We were assuming it is not necessary to include sequences of the amplicons since they were sequenced in the studies they were adapted from. However, sentence “PCR products were sequenced according to Eichmeier et al. [24] in order to validate the amplicon sequences.” was added to new lines 165-166 and 197-198 and amplicon sequences with aligned corresponding primer pair and probe for both the standard PCR and nested qPCR were added to appendix A as Figure A2 and Figure A3.
Question: line 190-201 – This is not necessary for this study, may be worth considering in the future study Gibbs and Gower method is an alternative option for quantifying the pathogen incidence in the seed or soil. (link: https://doi.org/10.1094/PDIS-03-18-0503-RE).
Answer: Thank you very much for your recommendation. We would surely consider this alternative approach of quantifying of the pathogen, as it seems less complicated and more suitable for this type of study.
Reviewer 2 Report
The authors present a study in which they develop a nested-PCR study to detect Xanthomonas spp. then attempt to leverage this to resolve questions about the persistence of these mircrobes in soil. However, the paper seems at odds with itself in several places with the latter half of the project seeming like it was added after the fact, rather than as an original goal of the method. While interesting, there are several critical issues that need to be addressed before this manuscript should be considered for publication:
- The title of the paper focuses on the field study but the abstract focuses on the nested PCR detection method. Which one is the primary focus of the paper? This 'split personality' issue is present throughout the document and needs to be addressed.
- Figure 2 shows data for the nested-PCR assay but the comparison to regular PCR data, which is arguably the only real conclusive statement from the paper is left in a supplemental figure. This needs to be restructured so they can be observed side by side. Even if there are methodological differences.
- Introduction needs to better reflect the majority of the paper which is about the field data an attempts to use this detection tool.
- It seems random that if the field study was part of the goal there was no spot sampling under different conditions to determine if culturable microbes were present. This feels hastily added as an attempt to refocus the paper
- The point of the final paragraph in the discussion is unclear. It seems to say this method is better than some but not better than others
- Shepherd's purse is indicated as a potential reservoir for Xcc, but no effort was made to determine if this was true. This weakens the statement the authors can make.
Author Response
Question: The title of the paper focuses on the field study but the abstract focuses on the nested PCR detection method. Which one is the primary focus of the paper? This 'split personality' issue is present throughout the document and needs to be addressed.
Answer: The primary focus of the paper is the field study. The whole paper was checked and edited to remove the split personality issue. Restructured the abstract (lines 20-25) as “…The aim of this study was to monitor persistence of Xcc in field soil in central Europe using a conventional PCR assay with hrpF primers and a two-step nested real-time PCR assay using Zur primers. The work has demonstrated that nested real-time PCR can be used to improve the analytical sensitivity for detection of Xcc in soil compared to conventional PCR, and that Xcc may persist in soil for up to two years following an infected brassica crop in central European climatic conditions.”.
Question: Figure 2 shows data for the nested-PCR assay but the comparison to regular PCR data, which is arguably the only real conclusive statement from the paper is left in a supplemental figure. This needs to be restructured so they can be observed side by side. Even if there are methodological differences.
Answer: Data from both assays, the conventional and the nested real-time PCR (Table A1 and Table A2) were merged together into a Table A1, so the data from two assays can be observed side by side. However, the table was left in Appendix A for now, as it became quite massive and fitting it into the main part of the paper might cause problems.
Question: Introduction needs to better reflect the majority of the paper which is about the field data an attempts to use this detection tool.
Answer: The introduction was restructured. A paragraph regarding survival of Xcc in soil has been added (lines 39-45) “Various studies have been made regarding the persistence of Xcc in soil, but most of them have focused on survival in association with crop debris. Thus it has been well established that Xcc can potentially survive in brassica crop debris in soil as long as debris remains visibly present [9,11-15]. On the other hand, Schaad and White [9] suggests that Xcc cannot survive in soil for more than 42 days in winter and 14 days in summer without the host plant debris.” and paragraph regarding ISTA seed testing “The current ISTA (International Seed Testing Association) method for detection of Xcc in seeds [14] combines dilution plating on selective media with either pathogenicity testing or PCR for confirmation of identity.” has been removed.
Question: It seems random that if the field study was part of the goal there was no spot sampling under different conditions to determine if culturable microbes were present. This feels hastily added as an attempt to refocus the paper.
Answer: The study was primarily focused on the field study (simple monitoring of Xcc in soil) by conventional and real-time PCR assays, which are commonly used in our laboratory. The real-time PCR was modified to a nested real-time for higher sensitivity of the assay, compared to the conventional PCR. The omitted attempts to isolate viable Xcc from soil was due to a flow in methodology. The isolation of viable microorganisms from soil is very difficult, due to high amount of microorganisms and due to a very competitive behaviour of many of them. We tried to cultivate Xcc on specific media from soil in time of the first sampling, but we were not able to isolate pure Xcc culture because of a huge number of microorganisms.
Question: The point of the final paragraph in the discussion is unclear. It seems to say this method is better than some but not better than others.
Answer: The contradictory final paragraph (lines 389-391) was restructured and merged with the previous paragraph as “It should also be noted that PCR is only the final step in the overall assay, thus whilst the PCR itself may be able to detect DNA equivalent to a single bacterial cell in a PCR tube, this represents only a small fraction of the original sample (in this case 0.01 g of soil), hence the overall detection limit of the nested PCR assay was 300 CFU·g-1 (the upper confidence limit of a negative result, P=0.95). Given the sensitivity of PCR we cannot be certain that all these positive results represent 'true positives', and it appears that the nested real-time PCR also introduced more variability into the results (presumably due factors such as the 60 % increase in cycle number, and increased number of operations e.g. pipetting), making direct interpretation of results more challenging. The precision of estimates was apparently improved by taking an MPN approach to interpretation of the data, and, with better overall assay design, e.g. by inclusion of multiple dilutions, or an efficient sequential assay design [39] this precision has potential for further improvement“ on lines 456-469. Lines 389-392 “…whereas a direct plating assay, where a larger fraction of the original sample is plated (e.g. 0.1 g) may have a lower detection limit of 30 CFU·g-1, and a bioassay such as the one used by Kocks [31] may be even more sensitive (0.34 CFU·g-1)”were removed. This part was not necessary, since according to Silva Junior et al. [33] and Kocks [31], the relatively low detected Xcc in our study has no epidemiological significance for following crops, therefor it is not necessary to mention other assays with lower detection limits.
Question: Shepherd's purse is indicated as a potential reservoir for Xcc, but no effort was made to determine if this was true. This weakens the statement the authors can make.
Answer: Lines 400-404 – The speculative statement was changed from “However, the cruciferous weed shepherd's purse (Capsella bursa-pastoris L.), which is known to be susceptible to Xcc race 4 [32], was observed growing in the experimental plot in both seasons. It is possible, therefore, that the soil Xcc population was maintained or recharged as a result of growth in or on the shepherd's purse during the experiment.” to “However, given that the experimental plot was left fallow and weeds were allowed to grow, the occasional occurrence of the cruciferous weed shepherd's purse (Capsella bursa-pastoris L.), which is known to be susceptible to Xcc race 4 [32], could also have affected the dynamics of detectable Xcc in the soil, in addition to initial population resulting from the infected crop.”. Currently, it really looks unfortunate that attempts were not made to isolate Xcc from the shepherd´s purse, but originally, we did not want to disturb the natural balance established in the experimental plot.
Round 2
Reviewer 2 Report
The revisions here do much to address my earlier concerns with regards to the text. Overall the paper reads more coherently and the relationship between the nested PCR experiment and the field study is more clear. There are a few examples of singular/plural errors throughout the document that ca easily be corrected during the proofing stage. Figure 3 is a little problematic because the means of several points cannot be seen (only error bars). An alternate format for this should be presented.
Author Response
Question: The revisions here do much to address my earlier concerns with regards to the text. Overall the paper reads more coherently and the relationship between the nested PCR experiment and the field study is more clear. There are a few examples of singular/plural errors throughout the document that can easily be corrected during the proofing stage. Figure 3 is a little problematic because the means of several points cannot be seen (only error bars). An alternate format for this should be presented.
Answer: We have thoroughly checked the grammar. Regarding the Figure 3, as explained in the text (lines 336-339), when all sub-samples are either positive or negative, it is not possible to obtain [and so plot] an MPN estimate, only a lower confidence limit (where all are positive) or an upper confidence limit (where all are negative). More precisely, in the case where all are negative, the estimate is actually zero, but the log of zero is undefined, and so cannot be plotted. We consider that it is better to be explicit that these values are not estimable, rather than present some arbitrary value. We have added an additional sentence in the caption for Figure 3 to make this clearer (lines 269-272).